# Antioxidant Activity of Quercetin-Containing Liposomes-in-Gel and Its Effect on Prevention and Treatment of Cutaneous Eczema

**DOI:** 10.3390/ph16081184

**Published:** 2023-08-21

**Authors:** Chang Liu, Xiaoman Cheng, Yifang Wu, Weifang Xu, Hongmei Xia, Ruoyang Jia, Yinyin Liu, Si Shen, Yinxiang Xu, Zhiqing Cheng

**Affiliations:** 1College of Pharmacy, Anhui University of Chinese Medicine, Hefei 230012, China; changliu@stu.ahtcm.edu.cn (C.L.); xmcheng@stu.ahtcm.edu.cn (X.C.); wuyifang@stu.ahtcm.edu.cn (Y.W.); pharamarcy_s@163.com (R.J.); jry3290912497@163.com (Y.L.); lyy183560335540516@163.com (S.S.); azhychzhiq@163.com (Z.C.); 2Anhui Province Key Laboratory of Pharmaceutical Preparation Technology and Application, Hefei 230012, China; 3Zhaoke (Hefei) Pharmaceutical Co., Ltd., Hefei 230088, China; centurygate@sina.com

**Keywords:** quercetin, liposomes-in-hydrogel, cutaneous eczema, antioxidant

## Abstract

Cutaneous eczema is a kind of skin disease is characterized by inflammation. The main manifestations are various types of dermatitis, eczema, and urticaria. There are usually complications such as erythema, blisters, and epidermal peeling. The quercetin might have a therapeutic effect on cutaneous eczema due to its favorable antioxidant activity and anti-inflammatory effects. Currently, there are few studies on transdermal administration of antioxidant drugs for the treatment of cutaneous eczema. The aim of this study was to prepare quercetin-containing liposomes-in-gel (QU-LG), its antioxidant properties were evaluated, and it was used in the skin of mice suffering from dermal eczema to see if it had preventive and therapeutic effects in an attempt to make it a new option for the treatment of cutaneous eczema. QU-LG was prepared by the injection method to form the quercetin-containing liposomes (QU-L) and evenly dispersed in the natural dissolution of carboxymethylcellulose sodium (1%, CMC-Na). The release of QU-LG across the dialysis membranes was up to 30% and clearance of 1,1-diphenyl-2-picrylhydrazyl (DPPH) was 65.16 ± 3.513%. In anti-oxidation assay QU-LG inhibited malondialdehyde (MDA) production in liver better than the commercially available drug dexamethasone acetate cream. Compared with untreated mice, mice treated with QU-LG showed a statistically significant reduction in dermatopathologic symptoms. The results suggested that QU-LG had good antioxidant activity in vivo and in vitro and could be used for the prevention and treatment of cutaneous eczema.

## 1. Introduction

With the rapid development of technology, people are stimulated by various sources of factors in their daily lives, and cutaneous eczema, as one of the most representative skin allergic diseases, has the characteristics of inflammation. In many countries, the prevalence of cutaneous eczema reaches 30% in children and 3% in adults [1,2,3,4]. The skin accounts for about 10–15% of body weight and is the largest organ in the body. [5]. When cutaneous eczema occurs, it can lead to many negative consequences, such as skin damage and other skin-based disorders. Cutaneous eczema is a peroxidative reaction of the skin caused by exposure to allergens. The degree of skin barrier damage and immune properties are usually determined by observing the number of keratinocytes in patients with cutaneous eczema. The main manifestations are various types of dermatitis, eczema, and urticaria. Complications such as erythema, blisters, and epidermal detachment usually occur [6,7]. In human skin, about 2% of oxygen consumption is converted to ROS (reactive oxygen species); the accumulation of ROS leads to an inflammatory response and accelerated skin oxidation [7,8,9]. Therefore, it is important to find ways to protect against skin damage caused by ROS.

Quercetin (QU, Figure 1), with its chemical name of 3,3′,4′, 5,7-pentahydroxyflavone (C_15_H_10_O_7_), is a member of the flavonol family and one of the most important dietary antioxidants. It is a natural polyphenol which is commonly found in various fruits and vegetables, such as citrus, potatoes, cabbage, peas, beans, apples, and ginkgo. QU has the effects of treating cardiovascular disease [10,11], anti-inflammatory [12,13], asthma [14], anti-nephrotoxicity [15], antioxidant [16], anti-cancer [17], and so on. Historical and epidemiological evidence shows that diet plans, including flavonoid such as quercetin, have positive health benefits, especially for the heart. Flavonoids have important medicinal value in the treatment of hypertension, diabetes, vascular disease, and cancer [18]. The characteristic of cutaneous eczema is an imbalance between Th1 and Th2, which is due to the presence of specific IgE reactions related to Th2 immune response. In cutaneous eczema, Th2 cells produce various cytokines, including IL-4, IL-5, IL-9, and IL-13, which play an important role in stimulating Th1 immune response. Quercetin can not only clear ROS, but can also inhibit cytokines produced by Th2 cells [19,20,21,22].

We prepared QU-containing liposomes (QU-L), QU-containing gels (QU-G), and QU-containing gel liposomes (QU-LG), and compared the effects of QU and its different formulations via the transdermal administration. In recent years, more and more drugs are choosing transdermal drug delivery system, so that the drug acts on the skin surface, the drug is absorbed through capillaries [23,24], and reaches the systemic blood circulation, thus, it acts locally or systemically. Transdermal administration is more convenient to use than other routes of administration such as oral and injection, avoidance of hepatic first-pass effect and gastrointestinal pH effect, and slow and controlled release. Liposomes-in-gel as drug delivery system could make the release of the drug slow down [25] and increased the bioavailability of the drug by utilizing the three-dimensional spatial network structure of the hydrogel, which leads to a better apposition of the dermal eczema to the skin surface.

## 2. Results

### 2.1. Standard Curve for QU Solution

The QU solution showed a linear correlation between absorbance and concentration in the range of 0.5–1.6 mg/mL with R^2^ = 0.9991, which is a good linear relationship (Figure 2). The QU solution was divided into five equal portions, the absorbance value at 374 nm was measured, and the relative standard deviation of precision was calculated RSD = 0.11%.

### 2.2. The Morphological Character and Physicochemical Properties of QU-L

Liposomes could repel each other and do not stick together to form clumps and precipitates [26] because of the same charge on the surface, and exist in a uniformly distributed form. The encapsulation efficiency of QU-L was as high as 85.39 ± 0.78%, indicating that liposomes encapsulate QU well and contain high drug content. Its potential and particle size were measured using the Malvern nanoparticle size potentiostat with the potential value of −32.5 ± 2.8 mV, particle size of 107 ± 10.23 nm. Polydispersity index was 0.243 ± 0.011 and the electron micrograph is shown in Figure 3A. QU is a flavonoid with four peaks between wavelengths 1624–1384 cm^−1^, an absorption peak between 800–680 cm^−1^, which is consistent with the characteristics of aromatic compounds a sharp absorption around 3462 cm^−1^, which is consistent with the characteristics of phenolic compounds, and a strong absorption at 1624 cm^−1^, which suggests the presence of C=O bonds. None of the quercetin liposomes detected the characteristic absorption peaks of quercetin at 3462 cm^−1^, 1624 cm^−1^ (Figure 3B), which indicated that quercetin was encapsulated in liposomes. In addition, computer software Discovery Studio 2016 was used to predict the interactions between formulation components. In the same space, there are Pi-Alkyl and Pi-Sigma forces between quercetin of different conformations and cholesterol, phospholipids, respectively, which are hydrophobic bonds (Figure 3C,D). Due to CMC-Na containing sodium ions, there are electrostatic forces of mutual attraction between quercetin and sodium ions, and weaker hydrogen bonds with carboxymethyl cellulose (Figure 3E). In addition, Alkyl interaction forces dominate between cholesterol molecules (Figure 3F).

### 2.3. Quality Evaluation Results of QU-LG

QU-G was prepared by using different proportions of gel matrix, and its viscosity, pH value, and recoverability were compared. The results showed that QU-G with 1% CMC-Na could maintain the viscosity of 1.5 ± 0.01 mpa·s. The pH value was stable at about 7.4, and the prepared hydrogel had a light-yellow appearance with good transparency. QU-G with higher contents of CMC-Na (3%, 4%, and 5%) showed excessive gel viscosity, semi-solid state, and matrix agglomeration in the gel. The gels prepared with contents lower than 1% were thin and did not meet our requirements for gels. The centrifugation test results showed no delamination and discoloration of QU-LG, which indicated that the stability of QU-LG was better. 

### 2.4. In Vitro Results of Antioxidation of QU, QU-L, QU-G, and QU-LG: Scavenging DPPH

DPPH is a free subcutaneous carcinogen centered on a nitrogen atom, and the unpaired electrons on the middle nitrogen atom cannot exert their proper electron-pairing effects due to the conjugation force and spatial potential between the three benzene rings. DPPH, as a stable free skin sensitizer, can trap other free skin sensitizers [27,28]. Therefore, DPPH is commonly used to observe whether the chemical reaction rate is slowing down. Since the center of DPPH has a strong absorption in the wavelength range of 400–600 nm, it appears as a dark purple color in solution and becomes colorless or light yellow after neutralization. Using this feature, the reaction process can be visually detected, and the color change can be used as an indicator of whether the reaction has free radicals. The initial number of free radicals can be obtained by recording the change in DPPH absorbance at 517 nm.

Free DPPH is soluble in ethanol and shows characteristic UV absorption at 517 nm [28]. When free radicals are added to the DPPH solution, the lone pairs interact with each other, thereby attenuating or disappearing the absorption of other free radicals around DPPH. Therefore, it is widely used in the determination of antioxidant activity. It was found that high concentrations of DPPH reacted better with quercetin, so it was used as a control compound for the antioxidant assay. The results (Figure 4) showed that the scavenging rate of quercetin on DPPH free radicals was curvilinearly related to the concentration in the range of 0–50 μg/mL, RE% = 0.625 (1 − exp(−0.1147 × *C*)), R^2^ = 0.9312, and the differences in scavenging rate results of QU group and QU-L group, QU-G group, and QU-LG group of different dosage forms were all statistically significant. 

### 2.5. Dialysis Membrane Release Rates of QU, QU-L, QU-G, and QU-LG

Different quercetin preparations showed different release behaviors across the dialysis membranes. According to the release curve (Figure 5), in terms of release rate results, QU-G > QU-L > QU-LG > QU-G. Theoretically, the release rate of QU should be the best. We hypothesize that the cause of this phenomenon is capillary action. QU soaks into the gap between the receiving and diffusing cells and then stays in the gap. It does not enter the reacceptance cell and therefore cannot be detected. QU-LG has the dual function of storing and slowing down the release of QU. The bilayer structure of phospholipids and the three-dimensional network of hydrogels made the release of QU slow down so that its R% was always lower than that of QU-L and QU-G solutions. By comparing the R% of QU-L and QU-G, we found that QU-L had a slower release effect on QU-G than QU-L, which indicated that the phospholipid bilayer structure had a longer-lasting release effect than the hydrogel network structure.

The release profiles were simulated using OriginPro 2021 software and the drug release kinetics were analyzed using commonly used models such as zero order, first order, Higuchi, Weibull, and Korsmeyer–Peppas. The equations, kinetic constants, and exponential parameters of the QU, QU-L, QU-G, and QU-LG curves fitted with different models are listed in Table 1. Using the WeibullCDF model, the R^2^ was >0.98 for all three dosage forms QU, QU-L, and QU-LG, but the R^2^ was lower for QU-G. Under the first order model, the R^2^ was >0.96 for all four dosage forms. The Higuchi model and the Korsmeyer–Peppas model gave lower R^2^ values for all the dosage forms, indicating that the time-accumulated release profiles of the drug did not follow the trend of the model, none of the four groups of dosage forms under the zero order model conformed to the fitted curves, and all the R^2^ were extremely small, hence the zero order model was excluded.

### 2.6. The Results of Applying QU-LG to the Treatment of Mouse Skin of Cutaneous Eczema

By taking pictures of the skin on the back of the mice every day (see Table 2), it could be seen that the skin of the mice in the model group showed large skin lesions within 1–7 days, and the lesions gradually expanded in scope and became more obvious after 8 days, and the thickness of the lesions increased significantly. The skin of mice in the blank control group was smooth and neat without eczema such as erythematous ulcers. Due to the use of compound dexamethasone acetate cream, the skin eczema of mice in the positive group gradually disappeared after the 13th day, and the skin eczema in the QU and QU-G groups still had a small amount of erythema on the 13th day, whereas the QU-L and QU-LG groups had returned to normal.

The splenic index is an organ coefficient that is the ratio of organ weight to body weight in experimental animals [29]. Under normal conditions, the splenic index is stable within a certain range. The value increases when the animal organ undergoes edema, inflammation, hyperplasia, and other diseases, and decreases when the animal organ undergoes lesions and degenerative diseases [30]. As can be seen from Table 3, the spleen index of mice in the model group was significantly higher than that of the blank and positive groups, and the blank group was the lowest. The spleen index of the groups of mice treated with quercetin and related preparations was lower than that of the positive group, among which the spleen index of the liposomes group was the smallest, which, to a certain extent, reflected that quercetin had a better therapeutic effect on cutaneous eczema. It suggested that the mice treated with QU, QU-L, QU-G, and QU-LG could inhibit the cutaneous eczema with the inflammation and swelling of the spleen. 

The morbidity of the mice in each group was scored (Table 4). Compared with the model group, mice in the treatment group had less wrinkled skin with eczema, smaller areas of erythema and ulcers, and milder symptoms. The average skin thickness of the mice in each group was measured. Blank control group: 0.52 ± 0.05 mm, model group: 0.98 ± 0.12 mm, almost twice as much as the blank control group (* *p* < 0.05), and the difference was statistically significant. Skin thickness in the QU, QU-L, QU-G, and QU-LG treatment groups was less than that in the model group and greater than that in the blank control group.

The epidermis of mice was paraffin-embedded, stained with hematoxylin eosin, and observed microscopically (Figure 6). Comparison of the dark purple part of the epidermis of the sections could be obtained by observing that the epidermis of the mice in the model group was obviously swollen, and the thickness of the epidermis of the positive group and the treatment group was reduced to a certain extent.

### 2.7. Results of MDA Experiments after Applying QU-LG to the Treatment of Mice with Cutaneous Eczema

MDA is a product of lipid peroxidation, which has a toxic effect on cells and is produced in two ways in the human body: firstly, arachidonic acid (AA) and other acid-like substances are produced by enzymatic reaction metamorphosis; secondly, polyunsaturated fatty acids (PUFA) are produced by non-enzymatic oxidative metamorphosis [31], which is used as one of the indicators of the degree of cellular damage. When the body is subjected to oxidative damage, a large amount of ROS accumulated in the body will disrupt the structure and function of cellular membranes, change the permeability, and make the cellular membranes undergo lipid peroxidation to produce MDA (Figure 7) [32], affecting normal cellular physiological and biochemical responses. The reaction between MDA and TBA is shown in Figure 8, and the reaction product has a maximum absorption at 532 nm.

In the MDA experiments (Figure 9), the results were essentially the same for both skin homogenates and liver homogenates. Absorbance in the model group > positive group > blank group, and the absorbance of quercetin and its preparations was within the range of theoretical values from the blank group to the model group. In liver homogenate, the absorbance of QU-L was the smallest, while in skin homogenate, the absorbance of QU-LG was the smallest, which might be related to the transdermal properties, liposome can well-penetrate the mouse skin to reach the body to exert therapeutic effects due to its structure and good biocompatibility, while the skin permeability of the hydrogel was worse. Therefore, in future clinical treatments, it can be considered to choose according to different diseases to achieve the best therapeutic effect.

## 3. Discussion

Quercetin has two hydroxyl groups in the A ring, which belongs to the hydroxyphenol structure, two hydroxyl groups in the B ring, which belongs to the odi-hydroxybenzene structure, and one hydroxyl group in the C ring, which belongs to the enol structure (Figure 1). The five hydroxyl groups of quercetin can quench free radicals and form stable phenoxy groups. Quercetin has low solubility in conventional solvents due to its structural features and leads to low bioavailability, thus limiting its clinical application [33]. Quercetin is modified by optimizing the structure and formulation to form its derivatives and related formulations in order to improve its bioavailability in the human body. In recent years, quercetin has also been shown to have good efficacy in the treatment of rheumatoid arthritis and interstitial lung fibrosis [34,35]. Therefore, there is an urgent need to develop its suitable dosage form [19,36].

In the proper formulation, quercetin has excellent efficacy in fighting against the inflammatory reactions on the skin. Inflammation is a multifactorial process involving the release of myeloperoxidase (MPO) by neutrophils and monocytes and for cutaneous eczema in the production of reactive oxygen species (ROS) associated with skin damage [37]. QU has excellent dual effects—coping with reactive oxygen species and increasing fibroblast proliferation. Fibroblasts are located in the dermis and are responsible for synthesizing collagen, glycoprotein, glycosaminoglycan, and other elements of the extracellular matrix of connective tissue, providing mechanical support for the skin. Therefore, skin fibroblasts play a key role in wound healing—without them, the wound site cannot regenerate extracellular matrix, epidermal skin cells cannot proliferate at the wound site, and the skin cannot recover from the injury [38]. 

The sodium carboxymethylcellulose (CMC-Na) improves the fluidity of vesicle bilayer and temporarily reduces the barrier function of the stratum corneum, thus creating easier channels for highly mobile vesicles so that quercetin can enter the stratum corneum completely (at least partially) and diffuse downward to the epidermis and dermis. Here, they come into contact with fibroblasts and promote drug absorption, enabling them to exert their anti-inflammatory/antioxidant activities when it is necessary (i.e., mainly in the dermis).

The structural analysis of quercetin and CMC-Na revealed that quercetin molecules have many hydroxyl groups that can form hydrogen bonds with water, often in the form of negatively charged ions (OH^−1^) in the aqueous solution of inorganic compounds, whereas sodium CMC-Na, which is a macromolecule, also has a large number of hydroxyl groups interacting through hydrogen bonds. When the content of CMC-Na was low, quercetin could bring together through moderate hydrogen bonding and electrostatic interactions, thus reducing quercetin hydration. Within a certain range, the more content of CMC-Na, the stronger the hydrogen bonding and electrostatic interactions between CMC-Na molecules. The quercetin molecule is surrounded by a large amount of sodium carboxymethyl cellulose. The conformational stability of quercetin increased due to the strong electrostatic repulsion between the two molecules. 

Hydrogels are networks of hydrophilic polymer chains, sometimes in a colloidal state in an aqueous phase. Hydrophilic polymer chains are crosslinked to form three-dimensional solids. Due to the existence of internal crosslinking, the structural integrity of the hydrogel network will not be dissolved by high water content. Hydrogels have high water absorption (more than 90% water content) and are very similar to natural tissues and are flexible. As a sensitive “smart material”, hydrogels can encapsulate chemical systems [39,40]. When stimulated by external factors such as pH changes, the chemical system releases specific compounds into the environment. 

The anhydrous ethanol solution of quercetin is volatile, and it has strong irritation to the human body. After the volatilization of alcohol, quercetin dissolved in it will precipitate and is not suitable for clinical use. In recent years, the anti-tumor, antioxidant, and anti-inflammatory effects of quercetin have been increasingly recognized [41]. Many quercetin dosage forms have been studied. We hope to produce low-toxicity and efficient dosage forms. A liposome is an artificial membrane. Spherical liposomes with a diameter of 25 nm to 1000 nm are formed after mixing. Liposomes can be used for gene modification or drug preparation [42]. In biology, when amphiphilic molecules (e.g., phospholipids and sphingolipids) are dispersed in the aqueous phase, the hydrophobic tails of the molecules tend to cluster together to avoid the aqueous phase. Liposomes are characterized by targeting, lytic targeting, slow release, reduced drug toxicity, and improved stability. Liposomes are mainly prepared by injection, film dispersion, ultrasonic dispersion, and reverse evaporation [43]. QU-L was prepared with a 3:1 ratio of phospholipid and cholesterol by injection [44]. Liposomes are rapidly internalized into the cell interior [45], and the drugs encapsulated in the liposomes are released after the vesicles are disrupted by the action of lysosomal and endosomal enzymes. This mode shows higher efficiency.

Human skin is the largest organ of the body and is resistant to various environmental insults. However, various factors induce a transient increase in ROS and an imbalance in the endogenous antioxidant system, which can increase free skin eczema and inflammation levels, which in turn affects cellular processes. The “-OH” group on the benzene ring of quercetin binds to important amino acid residues in the active site of acetylcholinesterase (AChE) and butylcholinesterase (BChE), which are involved in oxidative properties [46,47]. It has a strong inhibitory effect on the activity of these two enzymes. It also inhibits lipid oxidation by preventing the conversion of reactive oxygen species (ROS) into reactive chemicals. Studies have shown that QU, QU-L, QU-Q, and QU-LQ scavenged reactive oxygen species and enhanced the resistance of cell membranes and mitochondria to reactive oxygen species damage, thus preventing oxidative damage to the skin surface by external factors [48,49]. In skin eczema, it also inhibits cell membrane migration and mitochondrial membrane depolarization. Thus, the intake of QU, QU-L, QU-Q, and QU-LQ might also inhibit this imbalance.

## 4. Materials and Methods

### 4.1. Materials and Instruments

Quercetin (99%) was from Shanghai Suyi Chemical Reagent Co., Ltd. (Shanghai, China), soybean phospholipid (99%) and cholesterol (99%) were purchased from Tianjin Guangfu Fine Chemical Research Institute, sodium hydroxide was purchased from Sinopharm Chemical Reagent Co., Ltd. (Shanghai, China), and FeSO_4_·7H_2_O was purchased from China National Pharmaceutical Chemical Reagent Co., Ltd. Trichloroacetic acid (TCA) was produced by Tianjin Damao Chemical Reagent Factory in China, 2,4-dinitrochlorobenzene (DNCB), 1,1-diphenyl-2-pyridinhydrazine (DPPH, 98%), and thiobarbituric acid (TBA) was provided by Shanghai Yuanye Biotechnology Co., Ltd. (Shanghai, China). All reagents were of AR grade, and all experiments were conducted using deionized distilled water. Franze diffusion cell was purchased from Kunshan Ultrasonic Instrument Co. (Suzhou, China).

### 4.2. Animals

Healthy Kunming female mice (20 ± 2 g) were purchased from the Animal Experimental Center of Anhui University of Chinese Medicine (Hefei, China). Animal experiments adhered to the guidelines approved by the Ethics Committee of Anhui University of Traditional Chinese Medicine (Hefei, China).

### 4.3. Preparation of Quercetin Solution

0.23 g of quercetin powder was accurately weighed, dissolved in phosphate buffer solution (PBS, pH = 7.4), and concentrated in a 10 mL volumetric flask to obtain a QU solution at a concentration of 0.6 mg/mL, and kept for reference. The QU was configured with 0.575, 0.766, 0.92, 1.15, 1.3, and 1.5 mg/mL, and the UV absorbance values were measured and plotted as a standard curve.

### 4.4. Preparation of Quercetin-Containing Liposomes (QU-L)

The injection method was used to prepare QU-L, choosing a mass ratio of phospholipids to cholesterol of 3:1. 0.60 g of phospholipids and 0.20 g of cholesterol were accurately weighed into a beaker, and 5 mL of ethanol was added, sealed with plastic wrap, and dissolved completely in an ultrasonic water bath (70 °C) instrument. After complete dissolution, 10 mL of QU solution (2.3 mg/mL) was slowly poured into the beaker and stirred with ultrasonic waves until complete mixing and dissolution and was left overnight to allow the complete evaporation of ethanol and obtained QU-L. Blank liposomes (B-L) were prepared by replacing QU using the same volume of PBS as in the above procedure.

Take 2.5 μL of the sample on the copper grid and adsorb it for 1 min, then remove the excess sample from the side with a filter paper, drop 2.5 μL of 2% Uranyl Acetate on to the copper grid and then remove the excess dye from the side with a filter paper, repeat the process, then drop 2.5 μL of 2% Uranyl Acetate on to the copper grid and stain the sample for 1 min, then remove the excess dye from the side with a filter paper, and then finally, use a bulb type desk lamp to copy the copper grid dry. The samples were mixed with a turbo meter before use and later placed under a 120 KV transmission electron microscope (FEI Tecnai G2 spirit) for observation.

With the help of FTIR (Nicolite N10, Thermo, China), 32 scans at 4000~400 cm^−1^ at a resolution of around 8 cm^−1^ were performed on the infrared spectra of B-L and QU-L. The strength and displacement of the vibrational bands of the FTIR spectral peaks were used to estimate whether the drug was encapsulated and the compatibility of the formulation components. Experiments were performed thrice on all samples. Additionally, with the help of Discovery Studio 2016, small molecules were first prepared, and then small molecules with different conformations were selected for intermolecular force prediction.

Dilute QU-L 10 times with PBS, take 4 mL, centrifuge it at 4500 rpm for 15 min, take the supernatant and measure the absorbance at 370 nm, then calculate the concentration of the drug in the unencapsulated liposome according to the standard curve. 

After 10-fold dilution of QU-L with ethanol, 4 mL was taken and the liposome membrane was dissolved by ultrasonic waves, the supernatant was taken as above, and the total drug concentration and the encapsulation efficiency (ER%) of QU-L were calculated according to the standard curve. The formula is as follows: Encapsulation Efficiency (*EE*%):(1)$$EE\%=\frac{\left[C_{w}-C_{n}\right]}{C_{w}}\times 100\%$$
where “*C_n_*” was the free drug concentration out of liposomes; “*C_w_*” was the whole drug concentration in liposomes.

### 4.5. Preparation of QU-Containing Hydrogel (QU-G)

For the screening of hydrogel formulations, the hydrogel matrix was prepared by dissolving 0.10 g of carboxymethyl cellulose (CMC-Na) and carbomer in 5 mL of PBS. 230 mg of quercetin was dissolved in 1 mL of PBS and dispersed in the two groups of hydrogel matrix, and then the PBS solution was fixed to 10 mL. The pH of the carbomer was adjusted to 7.4. It was allowed to stand overnight and waited for swelling. On the next day, it was observed that the QU-containing hydrogel with sodium carboxymethylcellulose as the gel matrix had good viscosity, while the carbomer group had almost no viscosity. Therefore, sodium carboxymethyl cellulose was selected as the gel matrix and then QU-G was prepared using CMC-Na as the gel matrix. 

Add 5 mL of the liquid to be measured in the Austrian viscometer (Figure 10), blow air from port 1 with a suction bulb, raise the surface of the liquid to be measured above the a line, start the timekeeping when the liquid surface reaches the line, stop the timekeeping when the liquid surface reaches the b line, record the time when the liquid to be measured flows from the a line to the b line, and calculate the viscosity with pure water as a control using the Formula (2), with “*h*” denoting pure water and “*s*” denoting the sample to be measured, and the unit of viscosity is mpa·s, denoted by “*ŋ*”; the unit of density is g/mL, denoted by “*ρ*”; the unit of density is g/mL, denoted by “ρ”; and the unit of density is “ρ”. “*s*” represents the sample to be tested, the unit of viscosity is mpa·s, expressed in “*ŋ*”; the unit of density is g/mL, expressed in “ρ”; the unit of time (t) is s. It is known that the density of pure water is 1.0 g/m L Viscosity at 25 °C is 0.8937 mpa. 5 mL of pure water formed a line through the b line time of 59 s. The viscosity of pure water is 0.8937 mpa·s at 25 °C.
(2)$$\eta_{s}=\frac{\rho_{s}t_{s}}{\rho_{h}t_{h}}\times \eta_{h}$$

### 4.6. Preparation of QU-Containing Liposomes-Hydrogel (QU-LG)

Following the above procedure, using the injection method, 0.10 g of CMC-Na was added to 10 mL of QU-L and B-L, stirred rapidly until the particles were dissolved, and allowed to stand for 24 h to allow the particles to swell sufficiently to obtain QU-LG and B-LG.

### 4.7. In Vitro Antioxidation: Scavenging DPPH Free Radical (DPPH)

Blank group: 2 mL PBS + 1 mL DPPH solution was reacted in a PE tube and kept in the dark. Absorbance values were measured at 517 nm at 0.5 h, 1 h, 1.5 h, 2 h, 4 h, 6 h, and 8 h.

QU: PBS was replaced with 2 mL of quercetin solution of different concentrations (0.0035, 0.007, 0.014, 0.028, 0.056 mg/mL, respectively) in the blank group, and the other operations were the same. The standard curve was plotted.

QU-G: 2 mL of quercetin-loaded hydrogel and blank-loaded hydrogel were taken.

QU-L: 2 mL of quercetin liposome gel and blank liposome were taken.

QU-LG: 2 mL of quercetin liposome gel and blank liposome gel were taken.

Control group: no PBS was added, and other steps were the same as the blank group.

The calculation formula is as follows: (3)$$SE\%=\frac{A_{o}-\left(A_{s}-A_{c}\right)}{A_{o}}\times 100\%$$
where “*A_o_*” is the absorbance value of blank group; “*A_s_*” is the absorbance value of sample groups; “*A_c_*” is the absorbance value of control group.

### 4.8. In Vitro Release Rates of QU, QU-L, QU-G, and QU-LG across the Dialysis Membranes

The release rate (R%) of the samples was measured using a Franz diffusion cell (Figure 11). The Franz diffusion cell simulates a constant temperature and uniformly distributed liquid environment in an organism by means of a thermostatically circulating water system and constant speed magnetic stirring.

To measure the volume of the diffusion pool, the receiving cell was filled with PBS, the dialysis membrane was placed between the diffusion cell and the receiving cell, air bubbles were removed, and the device was placed on a magnetic stirrer. The temperature was set to 37 °C, and 1.0 mL of PBS, QU, BL, QU-L, B-G, QU-G, B-LG, and QU-LG were added to the supply cell at 5, 10, 20, 30 min, 1, 2, 3 …12, 24, 36, 48, 60, 72, and 84 h to remove 2 mL of sample from the receiver cell. Replenish the receiver cell with 2 mL of PBS after each sample.

### 4.9. Preliminary Study on the Preventive and Therapeutic Effects of QU-LG on the Skin of Mice with Cutaneous Eczema

The mice were acclimatized and fed for a few days and shaved off an area of 2 × 2 cm^2^ on their backs one day before the start of the experiment. Thereafter, the mice were weighed, and dorsal bare skin thickness was measured after each day of experimental manipulation.

Prepare 5% DNCB solution (dissolved in a mixture of acetone: olive oil = 3:1) and dilute to 0.5%. Mice were divided into 7 groups: blank, model, positive and QU, QU-G, QU-L, and QU-LG. A piece of bare skin of about 2 × 2 cm was exposed on the back of the mice, and 100 μL of 0.5% DNCB solution was applied to induce eczema on the skin for 2 days, while the blank group was applied with the same volume of acetone and olive oil vehicle. The above steps were repeated for sensitization 5 days later. The blank and model groups were treated with PBS, the positive control group was coated with equal volume of dexamethasone acetate cream, and the treatment groups were coated with QU, QU-L, QU-G, and QU-LG for 12 consecutive days, respectively. The mice were fasted for 12 h after the last treatment, and after execution on day 17, the dorsal skin, liver, and spleen were collected for subsequent experiments. The spleen index was calculated as the ratio of spleen to body weight [50,51].

On day 7, 10, 12, and 15, the severity of skin eczema was observed and categorized as none, mild, moderate, and severe, and scored according to the severity of bleeding/erythema, itching/deflaking, edema, and abrasion/erosion. Mean values were calculated by scoring assessments throughout the study. The thickness of the skin in millimeters was measured in the same area using vernier calipers.

Skin sections were stained with H&E stain and the epidermal status of the mice was observed microscopically [52].

Female mice were dissected [9], and the livers were quickly removed, rinsed repeatedly with 4 °C saline, blotted dry on filter paper, weighed, rinsed with 9-fold cold saline, and homogenized with a glass homogenizer. After centrifugation for 15 min at 4000 rpm, 10% of the liver homogenate was prepared by taking the supernatant and was stored in a refrigerator at 4 °C (The selected quercetin concentrations were 1.15, 1.53, 2.3, 4.6, and 9.2 mg/mL, respectively).

1 mL of 10% liver homogenate was cutaneously added into a series of 10 mL centrifuge tubes and divided into sample group, model group, and blank group. The sample group was cutaneously add with 100 μL. The blank group and model group were replaced by normal saline.

After mixing, let it stand for 5 min, then add VitC (10 mmol/L) and FeSO_4_·7H_2_O (10 mmol/L) 100 μL. The blank group was replaced by normal saline. After mixing, the sample group and model group were shaken in 37 °C constant temperature shaker for 1.5 h, and the blank group was stored at 4 °C.

Add 2 mL TCA with concentration of 10% into each centrifuge tube, mix well and let it stand for 5 min, add 1 mL TBA with concentration of 0.67% (0.67 g/100 mL), mix well, seal the tube mouth with preservative film, make a small hole on the preservative film, take a water bath at 95 °C for 40 min, cool with running water, centrifuge at 4000 rpm for 8 min, and absorb the supernatant.

The ultraviolet spectrophotometer was adjusted to zero with normal saline, and the absorbance value was measured at 532 nm.

The inhibition rate (*IR*%) was calculated according to the following equation:(4)$$IR\%=\frac{A_{m}-A_{s}}{A_{m}-A_{b}}*100\%$$
where “*A_m_*” is the absorbance value of positive model group; “*A_s_*” is the absorbance value of sample groups; “*A_b_*” is the absorbance value of negative blank group.

## 5. Conclusions

In conclusion, the injection method was used to encapsulate quercetin into liposomes and was evenly dispersed into sodium carboxymethylcellulose hydrogels to improve the bioavailability of quercetin and to enhance the efficiency of quercetin dermal delivery. Experiments showed that the quercetin-containing liposomes-in-gel had good skin stability and skin adhesion. In the DPPH assay, QU-LG showed the slow release and strong scavenging ability on free oxidized radicals. In the mouse skin eczema model, QU-LG significantly cured the epidermal eczema of the mice and inhibited the production of MDA in the skin and liver, which indirectly proved that QU could slow-release from liposomes-in-gel carrier and reach the place of ROS aggregation through the mucous membrane of the skin to scavenge the ROS and to alleviate the impact of oxidized free radicals on skin eczema. A drug delivery liposome gel that is portable and easy to use may help the subsequent development of new eczema drugs.

## Figures and Tables

**Figure 1 pharmaceuticals-16-01184-f001:**
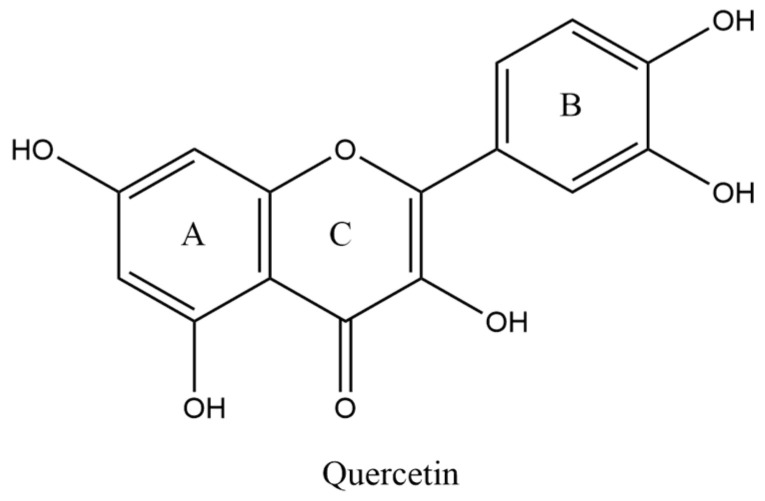
The structural formula of quercetin.

**Figure 2 pharmaceuticals-16-01184-f002:**
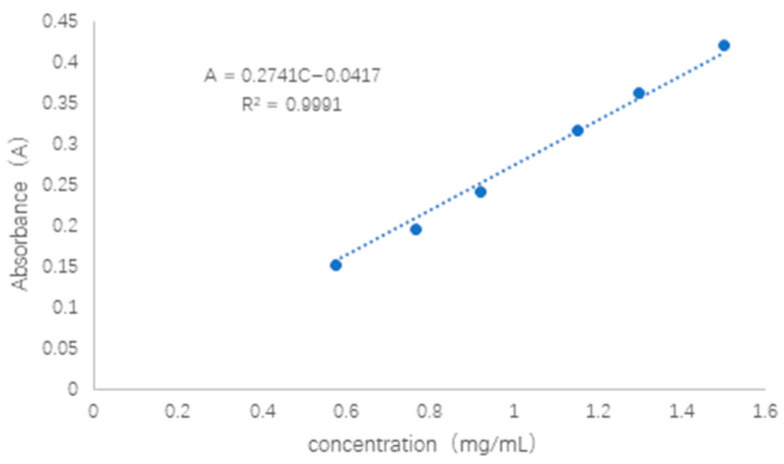
The standard curve of QU solution.

**Figure 3 pharmaceuticals-16-01184-f003:**
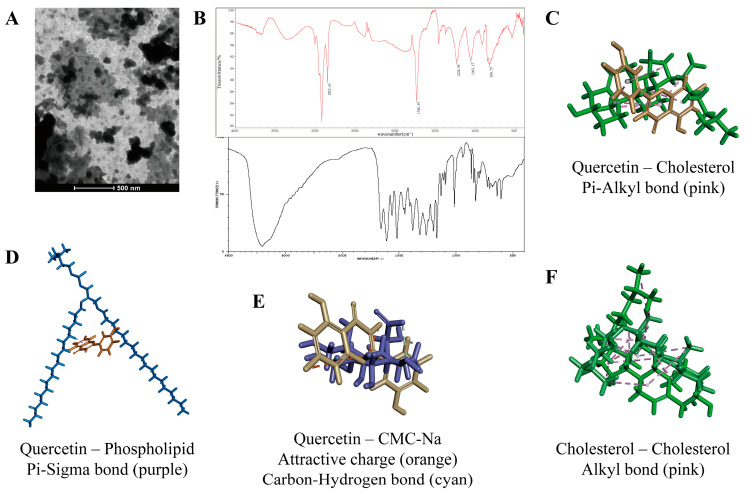
The characteristics and physicochemical properties of QU-L. (**A**) Electron micrograph of QU-L. (**B**) Infrared spectral profiles of QU-L (**top**) and QU (**bottom**, from SDBSWeb: https://sdbs.db.aist.go.jp/sdbs/cgi-bin/landingpage?sdbsno=2621, accessed on 25 July 2023). (**C**–**F**) Intermolecular interaction force results of QU with cholesterol, phospholipids, and CMC-Na and cholesterol with cholesterol predicted by Discovery Studio 2016.

**Figure 4 pharmaceuticals-16-01184-f004:**
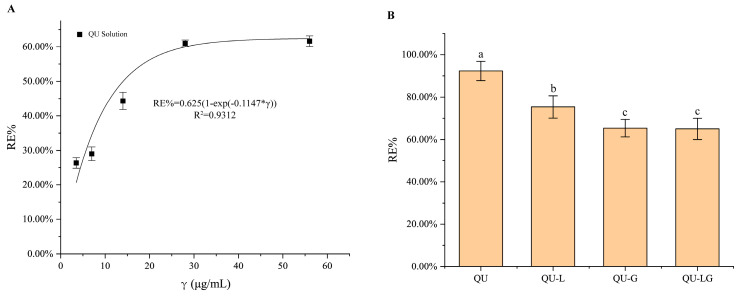
The results of DPPH free radical scavenging rate. (**A**) Different concentrations of QU. (**B**) The same concentration of QU, QU-L, QU-G, and QU-LG. Means with different letters (a–c) are significantly different (*p* < 0.05) via Duncan’s test.

**Figure 5 pharmaceuticals-16-01184-f005:**
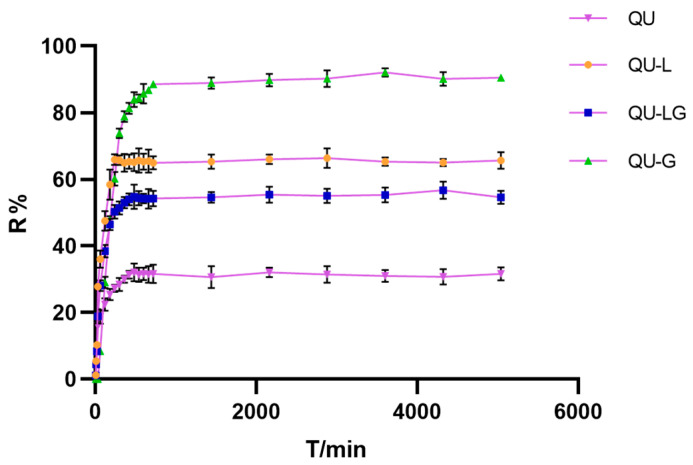
Experimental results of dialysis membrane release rate.

**Figure 6 pharmaceuticals-16-01184-f006:**
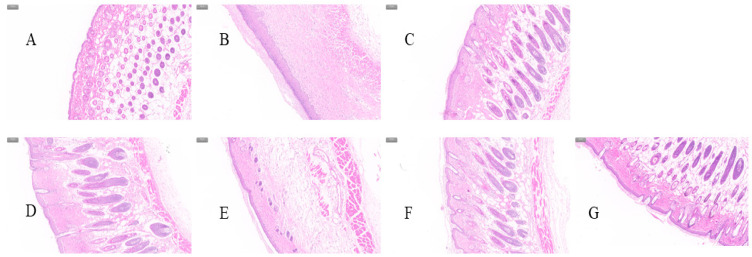
H&E-stained skin sections of mice with cutaneous eczema: (**A**) Blank group. (**B**) Model group. (**C**) Positive group. (**D**) QU. (**E**) QU-L. (**F**) QU-G. (**G**) QU-LG. (Scale: 100 µm).

**Figure 7 pharmaceuticals-16-01184-f007:**
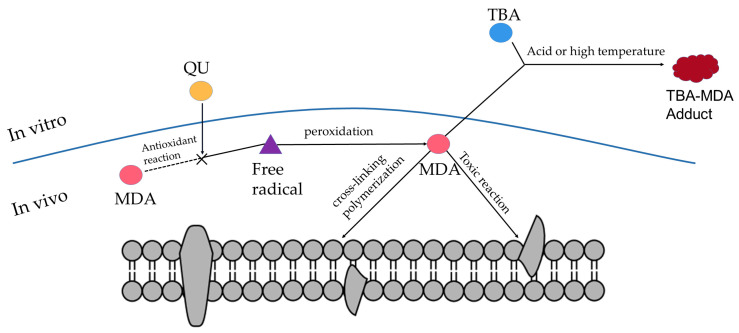
Schematic diagram of the TBA experiment.

**Figure 8 pharmaceuticals-16-01184-f008:**
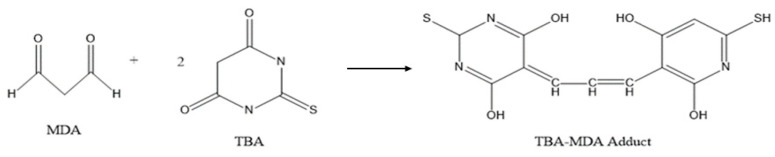
The product has the maximum absorption peak at 532 nm.

**Figure 9 pharmaceuticals-16-01184-f009:**
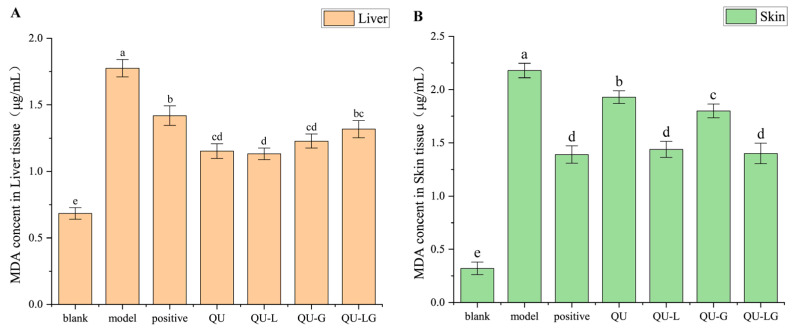
Results of MDA experiments in mice with cutaneous eczema, (**A**) liver, and (**B**) skin. Means with different letters (a–e) are significantly different (*p* < 0.05) via Duncan’s test.

**Figure 10 pharmaceuticals-16-01184-f010:**
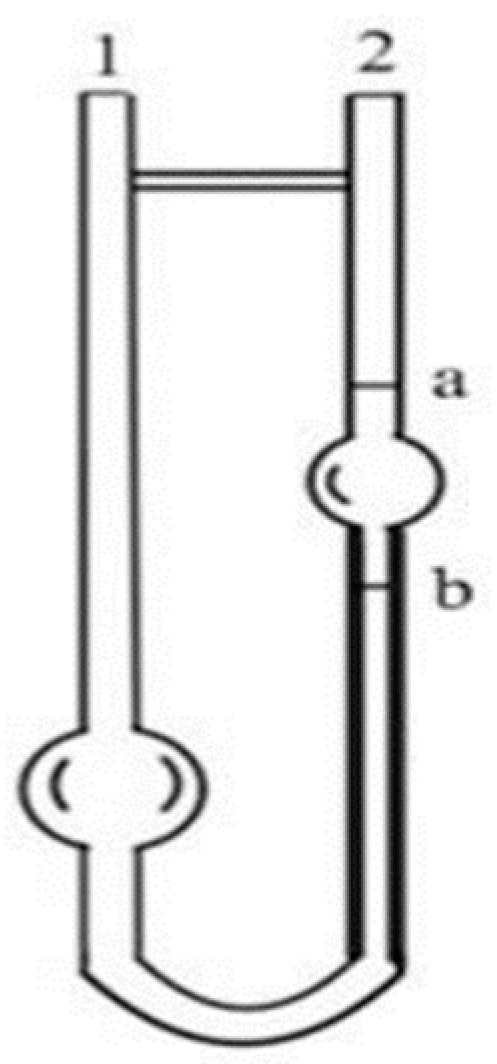
Austrian Viscometer. “1” is the inlet tube, “2” is the measuring tube, “a” and “b” are circular measuring lines.

**Figure 11 pharmaceuticals-16-01184-f011:**
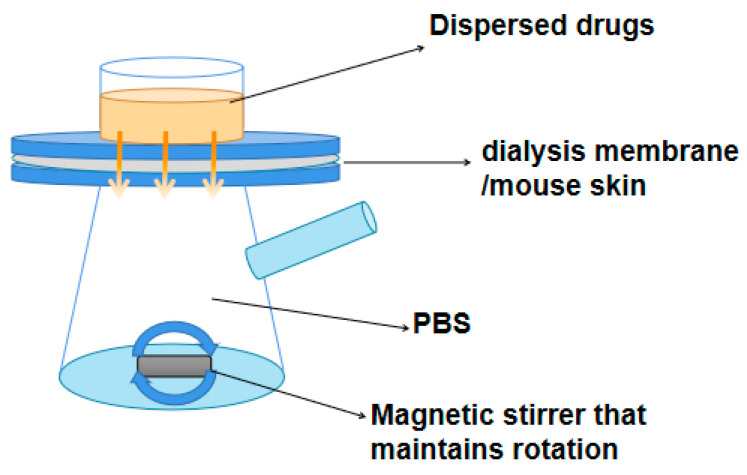
The diagram of Franz diffusion pool.

**Table 1 pharmaceuticals-16-01184-t001:** Simulation model of diffusion experiment across the dialysis membrane.

	QU-L	QU-G
First-order	$Q=65.70\left(1-exp\left(-0.0127t\right)\right)$ R^2^ = 0.9881	$Q=92.09\left(1-\exp\left(-0.0043t\right)\right)$ R^2^ = 0.9784
Higuchi	Q=0.64t^0.5+36.29 R^2^ = 0.3403	Q=1.30t^0.5+27.36 R^2^ = 0.5228
Weibull	Q=65.791−exp−0.014∗x−4.519^0.894 R^2^ = 0.9888	Q=299.871−exp−1.13∗10^−5x−5^0.271 R^2^ = 0.7184
Korsmeyer–Peppas	Q=20.34t^0.1602 R^2^ = 0.6201	Q=13.67t^0.2467 R^2^ = 0.6971
Zero order	Q=0.0061t+46.44 R^2^ = 0.1351	Q=0.0134t+46.56 R^2^ = 0.2758
	**QU-LG**	**QU**
First-order	$Q=54.69\left(1-\exp\left(-0.0109t\right)\right)$ R^2^ = 0.9946	$Q=30.76\left(1-exp\left(-0.0145t\right)\right)$ R^2^ = 0.9659
Higuchi	Q=0.58t^0.5+28.34 R^2^ = 0.3957	Q=2.94t^0.5+17.69 R^2^ = 0.3764
Weibull	Q=55.021−exp−0.012∗x−4.644^0.854 R^2^ = 0.9962	Q=30.611−exp−0.014∗x−4.74^0.643 R^2^ = 0.9862
Korsmeyer–Peppas	Q=15.51t^0.1717 R^2^ = 0.6657	Q=10.10t^0.1538 R^2^ = 0.6680
Zero order	Q=0.0057t+37.31 R^2^ = 0.1747	Q=0.0028t+22.28 R^2^ = 0.1594

**Table 2 pharmaceuticals-16-01184-t002:** Skin status of mice in each group within 16 days.

	Day1	Day 4	Day 7	Day 10	Day 13	Day 16
Blank	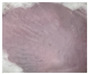	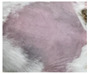	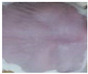	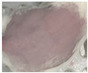	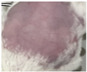	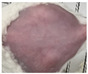
Model	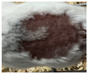	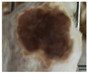	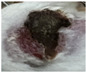	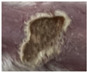	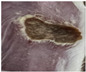	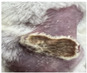
Positive	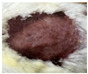	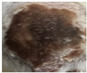	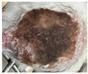	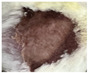	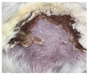	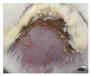
QU	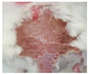	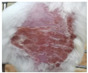	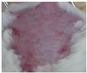	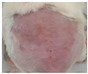	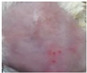	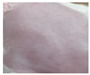
QU-L	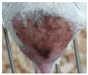	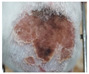	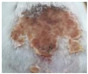	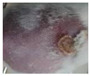	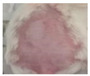	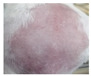
QU-G	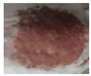	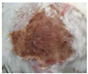	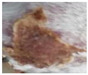	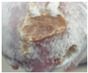	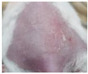	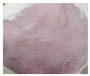
QU-LG	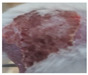	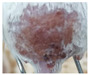	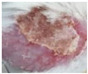	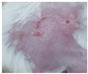	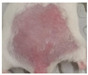	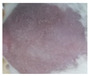

**Table 3 pharmaceuticals-16-01184-t003:** Mouse spleen index.

Group	Weight of Spleen(g)	Weight(g)	Splenic Index
Blank	0.101	28.88	0.3497
Model	0.094	23.46	0.4036
Positive	0.085	21.74	0.3919
QU	0.124	32.25	0.3838
QU-L	0.099	28.87	0.3460
QU-G	0.109	28.22	0.3876
QU-LG	0.110	28.7	0.3846

**Table 4 pharmaceuticals-16-01184-t004:** Skin appearance evaluation standard of mice.

Group	Appearance				
	Wrinkle	Roughness	Edema	Erythema	Ulceration
Blank	-	-	-	-	-
Model	+++	+++	+	+++	++
Positive	++	++	-	++	+
QU	+	+	+	+	+
QU-L	+	+	+	-	-
QU-G	+	+	-	+	-
QU-LG	+	+	-	-	-

“-” means none, “+” means mild, “++” means moderate, “+++” means severe.

## Data Availability

Data is contained within the article.

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
