# Peer review of "Antioxidant Activity of Quercetin-Containing Liposomes-in-Gel and Its Effect on Prevention and Treatment of Cutaneous Eczema"

_pharmaceuticals, 2023, doi:10.3390/ph16081184_

Round 1
Reviewer 1 Report
Dear Author,
I would like to inform you that the article entitled “Antioxidant activity of quercetin-containing liposomes-in-gel and its effect on prevention and treatment of cutaneous eczema” has been intensively reviewed and evaluated. The rationale of this study is outstanding but there were some major points that need to be intensively and carefully revised. After the completion of them, it could be a good candidate for literature.
Hereby I would like to present my comments:
Comment_1: As a general comment, please carefully use abbreviations and sustain its consistency. (L46: please add explanation, then continue, L49: Quercetin was abbreviated as Qu, then whole manuscript includes QU please keep consistency, L55, L61, L64, L112, L114, L125…etc.: Sustain your quercetin abbreviation with QU, L308, L314, L323…etc.: sodium carboxymethylcellulose.
Comment_2: Please check and organize the word in L71 “…dermal routes…”. Dermal is not convenient in this case.
Comment_3: The physicochemical characterization studies were not included in the manuscript. This was a major concern of pharmaceutical formulation development. In this perspective, the authors should demonstrate the physicochemical interactions of formulation ingredients by using Fourier Transform Infrared Spectroscopy (FTIR). Then the results should be thoroughly discussed.
Comment_4: The particle size, polydispersity index and zeta potential of the formulation are critical quality attributes for pharmaceutical development. Thus, the authors should provide these parameters as a part of the study.
Comment_5: Another major concern is the lack of evidence of formation of liposomes. The researchers should provide data about the morphology of liposomes (Polarized light microscopy, electron microscopy or another suitable techniques could be applied).
Comment_6: (Determination method of quercetin) The quantification of quercetin is another major issue. Please specify the quantification method (HPLC or UV) for the in vitro release studies and mention the Limit of Detection and Limit of Quantification values as a regular results of validation study.
Comment_7: (In L85: …different proportions of gel matrices…) Please share your viscosity values (the measurement method (in the method section) and viscosity values) and indicate your formulation chart (concentration versus viscosities… etc.).
Comment_8: (L140-142) Why the authors exclude zero order release kinetics, please explain in the manuscript or add and compare the results by using the zero-order release kinetics.
Comment_9: (L162-163) Please add citation for the definition.
Comment_10: (L194, Table 2) Please organize the table, the columns were mixed.
Comment_11: (L229, Table 3) please control the “black”, could it be “blank)
Comment_12: (L239-243) Please use a histopathological scoring system for the comparison. Then apply statistical analysis.
Comment_13: (L280-282) Please add a suitable citation for the sentence.
Comment_14: (L284-285) Please add a suitable citation for the sentence.
Comment_15: (As a suggestion) The sentence in lines L323 and 324 could be redundant or should be moved to another part of the paragraph.
Comment_16: (L327-328) Please add a suitable citation for the sentence.
Comment_17: (L336-337) Please add a suitable citation for the sentence.
Comment_18: (L338) Please check your reference which is not suitable for your explanation.
Comment_19: (L349-352) Please add at least two suitable citations for the sentence.
Comment_20: (L367, 4.2 Animals section) Please provide an approval number and date, then also provide the ethical approval form as a supplementary data.
Comment_21: (L378-381) How did you determine this formulation ratio? Please share your formulation chart or formulation design. Alternatively, if there was a previous study which include QU formulation please cite.
Comment 22: (L426) Please indicate the brand name or supplier of your Franz Diffusion Cell. (If it was handmade, please ignore this request).
Comment_23: (L437-441) please indicate the volume of receptor medium. Also mention the solubility of your active substance in your release medium. Then, indicate the release study was performed under sink conditions or not.
Comment_24: (L446-449) Please provide the data about approximate weight and genus name of mice.
Comment_25: (L451) please clarify the mice groups treatment groups were increased the group number (…Mice were divided into 4 groups…)
Comment_26: (L463-467) Please give details about the scoring system, if possible, add citations.
Comment_27: (L471) Please check your citation (Wu Y. et al [43].)
Best regards.
Dear Author,,
I believe that your article is sufficient in terms of language. I recommend correcting some minor spelling errors.
Best regards.
Author Response
Response to Reviewer 1
Dear Reviewer:
On behalf of my coauthors, we sincerely thank you for giving us an opportunity to revise the manuscript. We have carefully studied these opinions and revised them, hoping to get approval. We adopted your valuable suggestions and revised the article. For the parts of the manuscript that need to be improved, we have used red marks in the paper.
Point 1: As a general comment, please carefully use abbreviations and sustain its consistency. (L46: please add explanation, then continue, L49: Quercetin was abbreviated as Qu, then whole manuscript includes QU please keep consistency, L55, L61, L64, L112, L114, L125…etc.: Sustain your quercetin abbreviation with QU, L308, L314, L323…etc.: sodium carboxymethylcellulose.
Response 1: Thank you to the reviewers for their suggestions. Following your suggestion, we have added a note to "ROS" in L46. Change Qu to QU in L50, L53, ..., and L306... was changed from “carboxymethylcellulose” to “CMC-Na”.
Point 2: Please check and organize the word in L71 “…dermal routes…”. Dermal is not convenient in this case.
Response 2: Thank you to the reviewers for their suggestions. We have modified the formulation at L72.
Point 3: The physicochemical characterization studies were not included in the manuscript. This was a major concern of pharmaceutical formulation development. In this perspective, the authors should demonstrate the physicochemical interactions of formulation ingredients by using Fourier Transform Infrared Spectroscopy (FTIR). Then the results should be thoroughly discussed.
Response 3: Thank you to the reviewers for their suggestions. We are very sorry that we did not use FTIR to demonstrate the physicochemical interactions of the formulation components, which we will add in subsequent experiments.
Point 4: The particle size, polydispersity index and zeta potential of the formulation are critical quality attributes for pharmaceutical development. Thus, the authors should provide these parameters as a part of the study.
Response 4: Thank you to the reviewers for their suggestions. We have added data of particle size, zeta potential and polydispersity index (PDI) to L90-L93 in the manuscript.
Point 5: Another major concern is the lack of evidence of formation of liposomes. The researchers should provide data about the morphology of liposomes (Polarized light microscopy, electron microscopy or another suitable techniques could be applied).
Response 5: Thank you to the reviewers for their suggestions. We have added to the manuscript the observations of QU-L under electron microscopy. (Figure 3)
Point 6: (Determination method of quercetin) The quantification of quercetin is another major issue. Please specify the quantification method (HPLC or UV) for the in vitro release studies and mention the Limit of Detection and Limit of Quantification values as a regular results of validation study.
Response 6: We thank the reviewers for their suggestions. We have added standard curves L80-L82 for QU solutions to the manuscript and subsequent experiments to convert absorbance to concentration via standard curve equations, and we have added steps to plot standard curves in L374-L376.
Point 7:(In L85: …different proportions of gel matrices…) Please share your viscosity values (the measurement method (in the method section) and viscosity values) and indicate your formulation chart (concentration versus viscosities… etc.).
Response 7: Thank you to the reviewers for their suggestions. We tested the viscosity of the prepared QU-G with an Austrian viscometer. Relevant content has been added to manuscripts L99, L409-L422.Regarding the concentration of CMC-Na, there is a precedent of using 1% concentration in reference [9]. When the concentration is less than 1%, the prepared QU-G has no obvious difference with water, and when the concentration is 3%, 4%, and 5%, the agglomeration is visible to the naked eye, which is not suitable for our needs, and therefore we did not test the viscosity.
Point 8: (L140-142) Why the authors exclude zero order release kinetics, please explain in the manuscript or add and compare the results by using the zero-order release kinetics.
Response 8: Thank you to the reviewers for their suggestions. Our fitting results show that the zero-order release kinetics do not match any of our dosage forms and are therefore excluded, and the results of the zero-order release kinetics fitting have now been added to Table 1.
Point 9: (L162-163) Please add citation for the definition.
Response 9: Thank you to the reviewers for their suggestions. We have added citations to L173 [29].
Point 10: (L194, Table 2) Please organize the table, the columns were mixed.
Response 10: Thank you to the reviewers for their suggestions. We've tweaked Table 2 and can now more visually represent the changes in mouse skin over 16 days.
Point 11: (L229, Table 3) please control the “black”, could it be “blank)
Response 11: Thank you to the reviewers for their suggestions. We've changed "black" to "blank" in Table 3.
Point 12: (L239-243) Please use a histopathological scoring system for the comparison. Then apply statistical analysis.
Response 12: Thank you to the reviewers for their suggestions. We performed HE staining of the skin to show the skin healing in different groups of mice at the cellular level, and did not explore in the direction of pathology, thank you very much for your interest in the results of our experiments in terms of pathology, which will be the direction of our experiments in the future.
Point 13: (L280-282) Please add a suitable citation for the sentence.
Response 13: Thank you to the reviewers for their suggestions. We have added citations to L280 [33].
Point 14: (L284-285) Please add a suitable citation for the sentence.
Response 14: Thank you to the reviewers for their suggestions. We have added citations to L283 [34,35].
Point 15: (As a suggestion) The sentence in lines L323 and 324 could be redundant or should be moved to another part of the paragraph.
Response 15: Thank you to the reviewers for their suggestions. We have removed some of the statements.
Point 16: (L327-328) Please add a suitable citation for the sentence.
Response 16: Thank you to the reviewers for their suggestions. We have added citations to L325 [41].
Point 17: (L336-337) Please add a suitable citation for the sentence.
Response 17: Thank you to the reviewers for their suggestions. We have added citations to L334 [43].
Point 18: (L338) Please check your reference which is not suitable for your explanation.
Response 18: Thank you to the reviewers for their suggestions. We have added citations to L335 [45].
Point 19: (L349-352) Please add at least two suitable citations for the sentence.
Response 19: Thank you to the reviewers for their suggestions. We have added citations to L349 [48,49].
Point 20: (L367, 4.2 Animals section) Please provide an approval number and date, then also provide the ethical approval form as a supplementary data.
Response 20: Thank you to the reviewers for their suggestions. We have added animal ethics numbers to the manuscript.
Point 21: (L378-381) How did you determine this formulation ratio? Please share your formulation chart or formulation design. Alternatively, if there was a previous study which include QU formulation please cite.
Response 21: Thank you to the reviewers for their suggestions. The liposome preparation recipes in the manuscript were derived from previous experimental comparisons by our group and are mentioned in the published literature [9].
Point 22: (L426) Please indicate the brand name or supplier of your Franz Diffusion Cell. (If it was handmade, please ignore this request).
Response 22: Thank you to the reviewers for their suggestions. The Franze diffusion cell used in this group was purchased from Kunshan Ultrasonic Instrument Co. We have added relevant content to the manuscript.
Point 23: (L437-441) please indicate the volume of receptor medium. Also mention the solubility of your active substance in your release medium. Then, indicate the release study was performed under sink conditions or not.
Response 23: Thank you to the reviewers for their suggestions. The diffusion medium volume was measured for each receiving cell and was roughly 15.0 mL ± 1.0 mL. In the release study, we chose PBS buffer as the diffusion medium because PBS buffer pH is 7.4, which is close to the pH of body fluids, and quercetin solubility in PBS can reach 9.2 mg/ml, which may be due to the fact that quercetin forms salts with the metal ions in PBS ions, which increased the solubility. For the release under settling conditions, we are very sorry that we didn't carry out any experiments in this area, this will be the next experiments we need to add.
Point 24: (L446-449) Please provide the data about approximate weight and genus name of mice.
Response 24: Thank you to the reviewers for their suggestions. We used Kunming mice weighing in at 20 plus or minus 2g. It is stated in 4.2 of the manuscript
Point 25: (L451) please clarify the mice groups treatment groups were increased the group number (…Mice were divided into 4 groups…)
Response 25: Thank you to the reviewers for their suggestions. We have made changes at L468-L469.
Point 26: (L463-467) Please give details about the scoring system, if possible, add citations.
Response 26: Thank you to the reviewers for their suggestions. We thank the reviewers for their suggestions. The degree of cutaneous eczema was assessed by scoring five aspects of mouse skin: wrinkles, roughness, edema, erythema, and ulceration, as mentioned in the cited literature [9].
Point 27: (L471) Please check your citation (Wu Y. et al [43].)
Response 27: Thank you to the reviewers for their suggestions. We have made changes in L488.
Thank you again for your valuable comments on this manuscript.
Reviewer 2 Report
The aim of this study was a preparation of quercetin-containing liposomes-in-gel (QU-LG), evaluation of their antioxidant properties and their application to the skin of mice with the cutaneous eczema to to test their therapeutic properties. This manuscript includes all needed sections (introduction, materials and methods, results, discussion and conclusion), which are well written, but some errors need to be corrected before publishing:
1) The abbreviation ROS appears for the first time in the text (line 46) and its full name should be put there in brackets.
2) The abbreviations Qu and QU are used in the text for quercetin. Please choose one and use it throughout the manuscript.
3) The word the should be written with a lowercase letter after the comma (line 38).
4) Figure 2 and Figure 4 are not mentioned anywhere in the text. Please correct that.
5) Concentrations in µg/mL are mass concentrations, denoted by γ (not c). Please correct that in the manuscript.
6) The dependence of RE% on mass concentration (Figure 2A) as well as simulated release curves (Table 1) do not show a linear dependence. Please correct that in the text.
7) Variables (for example A) throughout the text are not in italics. Please go through the entire text in detail and correct this.
8) The word quercetin (line 290) is at the beginning of the sentence, so it should be capitalized.
9) Please correct word toojk (line 387)
10) In many cases there is no space between the variable and the measuring unit (For example in lines 398, 435, 437, 472…). Please go through the entire manuscript in detail and correct that.
Author Response
Response to Reviewer 2
Dear Reviewer:
On behalf of my coauthors, we sincerely thank you for giving us an opportunity to revise the manuscript. We have carefully studied these opinions and revised them, hoping to get approval. We adopted your valuable suggestions and revised the article. For the parts of the manuscript that need to be improved, we have used red marks in the paper.
Point 1: The abbreviation ROS appears for the first time in the text (line 46) and its full name should be put there in brackets.
Response 1: Thank you to the reviewers for their suggestions. Following your suggestion, we have added a note to "ROS" in L46.
Point 2: The abbreviations Qu and QU are used in the text for quercetin. Please choose one and use it throughout the manuscript.
Response 2: Thank you to the reviewers for their suggestions. We've made changes in the manuscript.
Point 3: The word the should be written with a lowercase letter after the comma (line 38).
Response 3: Thank you to the reviewers for their suggestions. We've made changes in L38.
Point 4: Figure 2 and Figure 4 are not mentioned anywhere in the text. Please correct that.
Response 4: Thank you to the reviewers for their suggestions. We have annotated the location of the illustrations in Figure. 3 (formerly Figure. 2) and Figure 5 (formerly Figure. 4) in the manuscript.
Point 5: Concentrations in µg/mL are mass concentrations, denoted by γ (not c). Please correct that in the manuscript.
Response 5: Thank you to the reviewers for their suggestions. We've made changes in the manuscript.
Point 6: The dependence of RE% on mass concentration (Figure 2A) as well as simulated release curves (Table 1) do not show a linear dependence. Please correct that in the text.
Response 6: Thank you to the reviewers for their suggestions. We've made changes in the manuscript.
Point 7: Variables (for example A) throughout the text are not in italics. Please go through the entire text in detail and correct this.
Response 7: Thank you to the reviewers for their suggestions. We've made changes in the manuscript.
Point 8: The word quercetin (line 290) is at the beginning of the sentence, so it should be capitalized.
Response 8: Thank you to the reviewers for their suggestions. We've made changes in L288.
Point 9: Please correct word toojk (line 387)
Response 9: Thank you to the reviewers for their suggestions. We've made changes in L387.
Point 10: In many cases there is no space between the variable and the measuring unit (For example in lines 398, 435, 437, 472…). Please go through the entire manuscript in detail and correct that.
Response 10: Thank you to the reviewers for their suggestions. We've made changes in the manuscript.
Thank you again for your valuable comments on this manuscript.
Round 2
Reviewer 1 Report
Dear Author,
Firstly, I would like to emphasize that the corrections you made during the previous revision process were largely successful and acceptable. However, I regret to note that certain major points have not been fully resolved. Due to the reasons outlined below, I am returning with a request for a second round of revision. Please make the amendments as described below without committing to any modifications for future or another studies. I appreciate your consideration of the previous corrections and wish you success.
Comment_1: As an integral part of characterization studies FTIR study and its interpretation has crucial importance for revealing chemical interactions of formulation components. (FTIR is an universal and quickest instrument for pharmaceutical R&D laboratories, thus the authors could achieve easily this request).
Comment_2: Please add the SEM imaging procedure in your materials and methods section.
Comment_3: There is a need of an analytical method validation study (It was considered that previous request was not clearly understood). Please investigate the literature about analytical method validation and also you could check ICH Q2 guideline). After that please give information about LOD LOQ and check the suitability and accuracy of your analysis method.
Comment_4: Please check and revise the name of Franz diffusion cells (not Franze).
Comment_5: The histopathological interpretations should be supported by a rational scoring system. (The researchers should have all the histopathological data (or images) in their databases, thus they could apply a histopathological scoring)
Best regards.
Dear Author,
I believe that your article is sufficient in terms of language. I recommend correcting some minor spelling errors.
Best regards.
Author Response
Response to Reviewer 1
Dear Reviewer:
We sincerely thank you for giving us an opportunity to revise the manuscript. We have carefully studied these opinions and revised them, hoping to get approval. We adopted your valuable suggestions and revised the article. We added reference and result comparison on the original basis to make the experimental results look more intuitive and reliable. With your valuable suggestions, some contents of the article have changed significantly. Therefore, the "Results" title of the article has been renamed as "Results and Discussion" title, and the "Conclusion" section has been added. We appreciate your valuable feedback, which we use to improve the quality of our manuscript. For the parts of the manuscript that need to be improved, we have used red marks in the paper.
Point 1: As an integral part of characterization studies FTIR study and its interpretation has crucial importance for revealing chemical interactions of formulation components. (FTIR is an universal and quickest instrument for pharmaceutical R&D laboratories, thus the authors could achieve easily this request).
Response 1: Thanks for the reviewer’s kind suggestion. We appreciate your offer to supplement the Fourier Transform Infrared studies for the interpretation of chemical interactions of the components of the formulation, in addition to the prediction of the sites of action between the components of the formulation using the computer software Discovery Studio 2016. We added the results and analysis of this section to L94-108 of the manuscript and the figure 3 in L109-113 of the manuscript. As shown in the figure below, quercetin is a flavonoid with four peaks between wavelengths 1624-1384 cm-1, an absorption peak between 800-680 cm-1, which is consistent with the characteristics of aromatic compounds; a sharp absorption around 3462 cm-1, which is consistent with the characteristics of phenolic compounds; and a strong absorption at 1624 cm-1, which suggests the presence of C=O bonds. None of the quercetin liposomes detected the characteristic absorption peaks of quercetin at 3462 cm-1, 1624 cm-1, which indicated that quercetin was encapsulated in liposomes. In addition, we used the DS 2016 software to predict the sites of action between the components of the formulation as well as the forces. In the same space, there are Pi-Alkyl and Pi-Sigma forces between quercetin of different conformations and cholesterol, phospholipids, respectively, which are hydrophobic bonds. Since CMC-Na contains sodium ions, there are electrostatic forces of mutual attraction between quercetin and sodium ions, and weaker hydrogen bonds with carboxymethyl cellulose. In addition, Alkyl interaction forces dominate between cholesterol molecules. Moreover, we have added methodological descriptions for infrared spectroscopy research and DS 2016 software prediction in L407-413 of the manuscript.
Figure 3. The characteristics and physicochemical properties of QU-L. (A) Electron micrograph of QU-L. (B) Infrared spectral profiles of QU-L (top) and QU (bottom, from SDBSWeb: https://sdbs.db.aist.go.jp (National Institute of Advanced Industrial Science and Technology, date of access)). (C-F) Inter-molecular interaction force results of QU with cholesterol, phospholipids, and CMC-Na and cho-lesterol with cholesterol predicted by Discovery Studio 2016.
Point 2: Please add the SEM imaging procedure in your materials and methods section.
Response 2: Thank you for the reviewer’s suggestions. We have added to L399-L406 of the manuscript.
Point 3: There is a need of an analytical method validation study (It was considered that previous request was not clearly understood). Please investigate the literature about analytical method validation and also you could check ICH Q2 guideline). After that please give information about LOD LOQ and check the suitability and accuracy of your analysis method.
Response 3: Thanks for the reviewer’s suggestion. We have added relevant content to manuscripts L80-L83.We used a UV spectrophotometer with a detection limit in the range of 200-600 nm, and the standard curve of quercetin that we plotted in the manuscript proved that quercetin has a linear relationship between absorbance and concentration at 374 nm in the range of 0.5-1.6 mg/ml, and the concentrations of the quercetin-containing preparations used in the manuscript were within this range, and 374 nm was within the detection limit, so we concluded that LOD and LOQ did not have an effect on our experimental results.
Point 4: Please check and revise the name of Franz diffusion cells (not Franze).
Response 4: Thanks for the reviewer’s careful suggestion. We've made changes in the manuscript.
Point 5: The histopathological interpretations should be supported by a rational scoring system. (The researchers should have all the histopathological data (or images) in their databases, thus they could apply a histopathological scoring)
Response 5: Thanks for the reviewer’s suggestion. The state of mouse skin eczema has been very clearly observed in the manuscript, and the skin thickness has been recorded accordingly. Our HE stained sections were observed using Caseviewer at the same magnification, and the HE sections were only used as an auxiliary proof of cytoplasmic thickening of mouse skin cells, which can be clearly seen to be different in thickness. Currently, histopathology is mostly used for the observation of tumor cell sections, so we do not think it is necessary to perform histopathological analysis on HE sections.
Thank you again for your valuable comments on this manuscript.

Round 3
Reviewer 1 Report
Dear Author,
First and foremost, I would like to congratulate you on the effort and dedication you have put into your work. Despite the shortcomings in the validation of the analytical method, I would like to express a positive viewpoint regarding this study of yours. Before delving into the analysis, I strongly recommend that you carefully read and apply the ICH Q2 guideline. Wishing you continued success.
Best regards,
Dear Author,
Your article has been found to be appropriate in terms of language.
Best regards.